# Lgr5 Does Not Vary Throughout the Menstrual Cycle in Endometriotic Human Eutopic Endometrium

**DOI:** 10.3390/ijms20010022

**Published:** 2018-12-21

**Authors:** Júlia Vallvé-Juanico, Cristian Barón, Elena Suárez-Salvador, Josep Castellví, Agustín Ballesteros, Antonio Gil-Moreno, Xavier Santamaria

**Affiliations:** 1Department of Reproductive Medicina, IVIRMA-Barcelona, 08017 Barcelona, Spain; julia.vallve@vhir.org (J.V.-J.); agustin.ballesteros@ivirma.com (A.B.); 2Group of Biomedical Research in Gynecology, Vall Hebron Research Institute (VHIR) and University Hospital, 08035 Barcelona, Spain; crbaronv@gmail.com (C.B.); antonioimma@yahoo.es (A.G.-M.); 3Department of Gynecology, Vall d’Hebron University Hospital, 08035 Barcelona, Spain; elenafeb@yahoo.es; 4Department of Pathology, Vall d’Hebron University Hospital, 08035 Barcelona, Spain; joscastellvi@vhebron.net; 5Universitat Autònoma de Barcelona, Bellaterra, 08193 Barcelona, Spain; 6Igenomix Foundation, Paterna, 46980 Valencia, Spain

**Keywords:** LGR5, endometrium, endometriosis, menstrual cycle, macrophages

## Abstract

Endometriosis is characterized by the abnormal presence of endometrium outside of the uterus, resulting in pelvic pain and infertility. The leucine-rich repeat-containing G protein-coupled receptor 5 (LGR5) has been postulated to be a marker of stem cells in the endometrium. However, LGR5^+^ cells have a macrophage-like phenotype in this tissue, so it is unclear what role LGR5^+^ cells actually play in the endometrium. Macrophages serve an important function in the endometrium to maintain fertility, while LGR5^+^ cells generally have a role in tumor progression and are involved in invasion in some cancers. We sought to determine whether LGR5^+^ cells vary across the menstrual cycle in women with endometriosis and whether there are implications for LGR5 in the aggressiveness of endometriosis and reproductive outcomes. We performed immunofluorescence, flow cytometry, and primary culture in vitro experiments on eutopic and ectopic endometrium from healthy and endometriosis patients and observed that neither LGR5^+^ cells nor LGR5 expression varied throughout the cycle. Interestingly, we observed that LGR5^+^ cell percentage overexpressing CD163 (anti-inflammatory marker) was higher in healthy endometrium, suggesting that in endometriosis, endometrium presents a more pro-inflammatory phenotype that likely leads to poor obstetric outcomes. We also observed higher levels of LGR5^+^ cells in ectopic lesions compared to eutopic endometrium and specifically in deep infiltrating endometriosis, indicating that LGR5 could be involved in progression and aggressiveness of the disease.

## 1. Introduction

Endometriosis is a chronic estrogen-dependent disease characterized by the presence of endometrial tissue outside the uterine cavity. Primary symptoms of the disease, which affects approximately 10% of reproductive age women, are acute pelvic pain and/or infertility/subfertility [1]. Endometrium, which is composed of stromal and epithelial compartments, contains many immune cells and is a very dynamic tissue that is tightly regulated by ovarian hormones. 

The leucine-rich repeat containing G protein-coupled receptor 5 (LGR5) is a seven transmembrane receptor described as a stem cell marker in a variety of tissues, including the small intestine and hair follicles [2,3,4]. LGR5 has been identified in endometrium [5,6], but its role in endometrial function is unclear. Recently, it was discovered that LGR5-positive cells (LGR5^+^) from healthy endometrium have a hematopoietic origin [7]. Approximately half of the population of LGR5^+^ cells found in the endometrium co-express CD45, a leukocyte marker, and CD163, a monocyte and macrophage specific marker, suggesting a myeloid nature to the cells. This expression pattern was also observed by our group in LGR5^+^ cells from eutopic endometrium from women with endometriosis [8]. Interestingly, LGR5^+^ cells seem to remain constant throughout the menstrual cycle in normal human eutopic endometrium [7,9], although it was recently described that LGR5 expression decreases during the secretory phase [6]. 

We previously described a special subset of LGR5^+^ cells that express unique genes in the eutopic endometrium of women with deep infiltrating endometriosis (DIE) compared to other types of endometriosis. These genes are related to immune system responses, hematological system development, and infertility [8]. Thus, we believe that LGR5^+^ cells could be implicated in aggressiveness and reproductive outcomes of the disease based on the function of these co-expressed genes. 

LGR5^+^ cells have a macrophage-like phenotype, and macrophages have been shown to increase in the secretory and menstrual phases of the menstrual cycle in normal eutopic endometrium [10]. However, this increase has not been observed in eutopic endometrium of women with endometriosis [11]. Macrophages have a role in embryo implantation failure and are involved in poor reproductive outcomes [12]. Moreover, this immune cell population is increased in the peritoneal fluid and endometriotic lesions of women with endometriosis [13]. However, variation of these types of cells throughout the menstrual cycle in women with endometriosis has not been investigated. We assessed whether LGR5^+^ cells vary throughout the menstrual cycle, along with the percentage of LGR5^+^ cells in ectopic versus eutopic endometrium, as these cells may have implications for reproductive outcomes in endometriosis.

## 2. Results

### 2.1. LGR5 Does Not Vary Throughout the Menstrual Cycle in Women with Endometriosis 

To determine whether LGR5 varies throughout the menstrual cycle, we used three experimental approaches: immunofluorescence, in vitro assay, and flow cytometry. We used endometrial biopsies and ectopic lesions in order to answer this question. The characteristics of the patients are shown in Table 1. 

#### 2.1.1. Immunofluorescence Analysis

Calculations of fluorescence intensity mean (FIM) for LGR5 throughout the menstrual cycle showed no significant differences between epithelial and stromal compartments in control endometrium and no significant differences in the case of endometriosis (Figure 1A). Additionally, there were no significant differences between phases of the menstrual cycle in either group (Figure 1B). However, when comparing LGR5 expression in the eutopic endometrium from healthy patients and endometriosis patients, we observed a statistically significant increase in LGR5 expression in proliferative, secretory, and menstrual phases in healthy women (Figure 1C). An example of the immunofluorescence in eutopic endometrium is shown in Figure 1D. 

#### 2.1.2. In vitro Studies

In vitro experiments using endometrial stromal fibroblasts (eSF) cells treated with estradiol (E_2_) for six days, and with E_2_ and progesterone (P_4_) for six more days, showed that primary culture cells mimicked proliferative and secretory phases of the menstrual cycle. Although non-significant, a tendency in increase in expression of CYR61, which increases during the proliferative phase, was observed and a tendency in increase of DKK1, which increases during the secretory phase, was also observed at days 6 and 12, respectively, in both groups when compared to the untreated control (Figure 2A). No differences in cell morphology were observed. In addition, no significant variation in LGR5 throughout the menstrual cycle was observed in control or endometriosis samples. Interestingly, a non-significant decrease in LGR5 was observed in the proliferative phase of the treatment group in control samples, while we observed a non-significant increase in the secretory phase of endometriosis samples (Figure 2B). 

#### 2.1.3. Flow Cytometry Analysis

In order to determine the variation throughout the cycle, we next performed flow cytometry analysis to determine the percentage of LGR5^+^ cells in eutopic and ectopic endometrium. We measured a mean of 5.9% LGR5^+^ cells (range: 2–7%) from each sample. In the preliminary study, no variation in percentage of LGR5^+^ cells was observed throughout the menstrual cycle in nine eutopic endometriosis samples. Interestingly, no variation in LGR5^+^ cells across the menstrual cycle was observed (Figure 3A). Moreover, no significant differences were found between LGR5^+^ cells in eutopic endometrium of women with endometriosis (differentiated by different types of disease) (Figure 3B) and control women (Figure 3C). However, comparison between ectopic lesions and eutopic endometrium in patients with endometriosis (ovarian and DIE) revealed a significant increase of LGR5^+^ cells in ectopic lesions (Figure 3D). These comparisons were only performed for ovarian endometriosis and DIE due to the lack of ectopic samples from pelvic endometriosis (*n* = 1) and adenomyosis (*n* = 2). A *t*-test showed that DIE lesions contained a significantly higher percentage of LGR5^+^ cells than ovarian lesions (Figure 3E). 

## 3. Discussion

The endometrium is a highly dynamic tissue that changes throughout the menstrual cycle. LGR5 is an interesting cellular marker in endometrial cells that has been shown not to vary throughout the cycle at RNA and protein levels in healthy endometrium [5,9]. However, a recent study reported that LGR5 is regulated by progesterone and showed that it does decrease in the secretory phase of the menstrual cycle and that there are progesterone binding sites in the promoter of LGR5 [6]. Interestingly, stromal fibroblasts show progesterone resistance in women with endometriosis [14]. Despite these controversial findings, the behavior of LGR5^+^ cells in eutopic and ectopic endometrium throughout the menstrual cycle had yet to be described in women with endometriosis. 

Only four studies on LGR5 have been performed in healthy human eutopic endometrium and, to our knowledge, this is only the second work to explore LGR5 in the eutopic endometrium of women with endometriosis—the first study was published by our group [8]. Our previous work demonstrates an abnormal epithelial phenotype in the stromal compartment of eutopic endometrium of women with endometriosis [8], similar to what has been observed in mice with induced endometriosis [15]. LGR5 is aberrantly co-expressed with cytokeratin (CK) or E-cadherin (ECAD) in endometriotic eutopic endometrium, but this co-localization is not found in healthy women, suggesting that eutopic LCR5^+^ cells could have different behavior in endometriosis compared to healthy tissue [8]. Interestingly, we observed this process in both follicular and secretory phases of the menstrual cycle. 

Cervelló et al. showed that LGR5^+^ cells in healthy eutopic endometrium present a monocyte-macrophage-like phenotype and, surprisingly, do not express any typical stem cell markers, indicating that these cells do not act as traditional stem cells in this tissue [7]. However, these cells do seem to be involved in stem cell niche modulation [7]. These results, together with those from our group, are comparable to our findings in patients with endometriosis, in which we also observed a predominant myeloid phenotype of these cells [8], indicating they are likely monocytes and macrophages. 

In the endometrium, macrophages make up approximately 10% of the total immune cell population [10,16,17,18], making them the second most abundant endometrial leukocyte population after T cells [12]. Macrophages comprise 1–2% of endometrial cells in the proliferative phase, 3–5% in the secretory phase, and 6–15% in the menstrual phase [10]. Depending on the activation state and surface markers, they are classified as either classically activated macrophages (Mϕ1) or alternatively activated macrophages (Mϕ2) [19]. Mϕ1 secrete pro-inflammatory factors, whereas Mϕ2 are involved in angiogenesis, anti-inflammatory processes, and coordination of tissue repair [19,20]. This plasticity in phenotype is due to environmental cues [21]. In normal endometrium, the majority of macrophages are CD163^+^CD14Low, which correspond to Mϕ2 [20,22]. Interestingly, LGR5^+^ cells overexpress CD163 in normal endometrium [7], suggesting an anti-inflammatory phenotype in this tissue. 

In accordance with other groups [5], we did not observe significant differences in LGR5^+^ cell percentages between epithelial and stromal compartments as measured by flow cytometry and gene expression after RNA-high-sequencing analysis [8]. In the present work, we also assessed differences between percentages of LGR5^+^ cells in eutopic endometrium in healthy women and women with endometriosis in both menstrual phases. We did not observe significant differences in LGR5^+^ cells throughout the cycle in control or endometriosis groups, as similarly observed by other authors in normal, healthy endometrium [5]. However, we did observe a significant increase in LGR5 in healthy endometrium as measured by immunofluorescence in all phases of the menstrual cycle compared to women with endometriosis. This suggests that healthy eutopic endometrium has a higher anti-inflammatory phenotype than endometriotic endometrium, where the percentage of LGR5^+^ cells is lower. We hypothesize that a lower percentage of LGR5^+^ cells would lead to a pro-inflammatory phenotype throughout the menstrual cycle in eutopic endometrium of diseased women, which may have a negative impact on reproductive outcomes. 

We did not find differences in LGR5^+^ cells between healthy and endometriotic endometrium by flow cytometry, although immunofluorescence did show a significant increase in LGR5 in healthy endometria in all phases of the menstrual cycle compared to endometriosis. This discrepancy may be due to: (1) different forms of sample processing (fresh tissue or FFPE endometrium); (2) different sample sizes, since in the flow cytometry study the control group represented only one third of total endometriosis samples; (3) use of two different antibodies for each technique; and/or (4) the FACS control group being comprised of egg donors. Egg donor women were stimulated with follicle stimulating hormone (FSH), which could produce differences in LGR5 expression. Immunofluorescence is likely more reliable in this case because there is no bias of sample processing and control group members were healthy women without stimulated endometrium. Moreover, the sample size is slightly larger in the immunofluorescence analysis. 

To avoid possible effects from FSH in the stimulated endometrium, when we mimicked the menstrual cycle in vitro, we grew cells in DMEM without phenol red and we treated cells with charcoal-stripped serum, which inhibits androgen receptor expression. CYR61 and DKK1 expression was measured to find out if the treatment was having any effect in terms of gene expression in endometrial stromal fibroblasts. In a previous in silico study performed by our group [8], we determined that these two markers were varying across the menstrual cycle in healthy human eutopic endometrium. CYR61 was overexpressed in proliferative phase and DKK1 in secretory phase. Although it is not significant, after the treatment with E_2_ (mimicking proliferative phase) and E_2_P_4_ (mimicking secretory phase) we observed an increase of CYR61 at day 6 and an increase of DKK1 at day 12 in both groups when compared to the control stromal fibroblasts, indicating that the treatment of E_2_ and P_4_ was having the desired effect on the primary cells. Interestingly, we observed a slight decrease (not statistically significant) in LGR5 expression in the secretory phase when E_2_P_4_ was added to the medium in the control patients in in vitro experiments (Figure 2B), in accordance with Tempest et al. [6], that states that LGR5 could be regulated by progesterone. In women with endometriosis, we also observed no significant variation throughout the menstrual cycle in the in vitro experiments, although a trend for increased marker in the secretory phase suggests that LGR5 could be regulated by progesterone, because it has progesterone binding sites in its promoter [6] and there is resistance to progesterone in women with endometriosis [23].

Although we are aware that the number of ectopic lesions obtained is low, we found our results interesting because the average percentage of LGR5^+^ cells was increased in endometriotic lesions compared to their matched eutopic endometrium. Additionally, DIE ectopic lesions displayed a higher average of LGR5^+^ cells than ovarian endometriosis. Our previous findings show that LGR5^+^ cells from eutopic endometrium of women with DIE present a special subset of LGR5^+^ cells [8] that could not only play a role in endometriosis in the eutopic endometrium, but also in the development of ectopic lesions and their aggressiveness. 

Other works have supported this idea since LGR5^+^ seems to be related to progression of different cancers, such as colon [24], papillary thyroid [25], breast [26], and ovarian cancers [27] by promoting epithelial ovarian cancer proliferation, metastasis, and epithelial–mesenchymal transition (EMT). Furthermore, LGR5 is overexpressed in ovarian cancer tissue compared to normal tissue [28]. Moreover, it is well known that patients with endometriosis have a higher risk of developing ovarian cancer [29]. These findings, together with our results, suggest that LGR5 could also be involved in the pathophysiology of endometriosis and its eventual progression to ovarian cancer. However, the role of LGR5 in ectopic lesions and its relation to the promotion of endometriosis and/or ovarian cancer is less understood. LGR5 should be further considered as a marker of ovarian endometriosis lesions, and its role in the development of ovarian cancer should be further studied.

In conclusion, this work shows that LGR5 does not vary across the menstrual cycle in healthy and endometriotic eutopic endometria at either RNA or protein levels using three different approaches: immunofluorescence, RT-qPCR, and flow cytometry. It seems that healthy eutopic endometria have more LGR5^+^ cells, suggesting that endometriotic eutopic endometria could have a pro-inflammatory phenotype that would lead to poorer obstetrical outcomes. Our results open a new window to study LGR5^+^ cells in ectopic lesions and discover their role in the pathophysiology and aggressiveness of endometriosis. 

## 4. Materials and Methods

### 4.1. Sample Collection

A total of 101 samples were obtained from 24 eutopic endometrium embedded in paraffin (formalin-fixed paraffin tissue, FFPE) from women with endometriosis and 24 from healthy women that were provided by the Department of Anatomical Pathology of the University Hospital Vall d’Hebron, Barcelona. From those samples, we obtained five slides from phases of the menstrual cycle (early proliferative, late proliferative, early secretory, late secretory) and four slides from menstrual phases to perform immunofluorescence assays. We also obtained 12 fresh eutopic endometrium from healthy egg donors, provided by IVIRMA-Barcelona (Barcelona, Spain), for flow cytometry assays. Four of these samples were also used for in vitro assays. We also collected 28 eutopic and 13 ectopic endometrium from women with endometriosis who underwent laparoscopy from the University Hospital of Vall d’Hebron, Barcelona. Ectopic tissue samples were categorized as follows: four samples were ovarian endometriosis, six were DIE, two were adenomyosis, and one was pelvic endometriosis. These samples were mostly used for flow cytometry assays, except for three samples that were used for in vitro assays. Different types of endometriosis were collected, and all patients and their designations are listed in Table 1. All patients signed a consent form, and use of the uterine specimens after surgery was approved by the Ethics Committee of Vall d’Hebron Research Institute, Barcelona, Spain (PR(AMI)410/2016) and by the ethics committee of IVI Barcelona S.L. (1611-BCN-080-XS) on 7 July 2017.

### 4.2. Immunofluorescence 

Three cuts of 3 µm each were made from all paraffin blocks. A xylene/ethanol circuit and 5 min in H_2_O was used to deparaffinize and rehydrate tissues. Slides were then blocked with 5% bovine serum albumin (BSA) and 5% normal goat serum (NGS) in 1× phosphate buffered saline (PBS) for 30 min. We stained tissue with anti-LGR5 antibodies at 1:30 dilution (anti-LGR5/Gpr49 (loop 2) rabbit, Abgent, San Diego, CA, USA) overnight (ON) at 4 °C. After three washes with 1× PBS, anti-rabbit Alexa647 (Invitrogen, Carlsbad, CA, USA) secondary antibody was added at 1:500 dilution and incubated for 45 min at room temperature. All antibody dilutions were made using 3% BSA solution with 3% NGS in 1× PBS. We used ProLong Gold antifade reagent with 6-diamino-2-phenylindole (DAPI; Invitrogen) to visualize nuclear DNA and mount the slides. As negative controls, we stained the tissue only with secondary antibody and used unstained tissue. Two independent observers using an OlympusBX61 (Tokyo, Japan) microscope evaluated the slides. We took photos from five different fields for each sample. Next, images were analyzed with ImageJ software K1.45 (GPL). We took photos for each sample and split color channels in red for LGR5 and blue for DAPI. We selected epithelial and stromal regions and calculated intensity (fluorescence intensity mean (FIM)) levels of LGR5 in relation to DAPI to compare epithelial versus stromal LGR5 expression. Data were analyzed with GraphPad Prism 8 software (GraphPad Software Inc., San Diego, CA, USA) using two-way ANOVA and Bonferroni tests. A *p*-value of <0.05 was considered statistically significant. Different red signal intensity in relation to blue was used to determine the expression level for each cell type.

### 4.3. Tissue Digestion and Primary Culture

Endometrial tissue (four samples from controls and three from DIE) was minced mechanically and digested with DMEM high glucose and collagenase (1 mg/mL). After digestion at 4 °C ON, samples were separated by gravity sedimentation, and single cells were filtered in a 40-µm mesh and washed with 1× PBS. Stromal cells were collected as they were filtered through the mesh. Epithelial cells remained in the mesh, so the mesh was washed upside down with 1× PBS to recover epithelial cells. Afterwards, epithelial cells were incubated with accutase to obtain singe cells. Pellets were washed with 1× PBS and resuspended with serum containing medium (SCM). 

Using endometrial stromal fibroblast (eSF) primary cell cultures, an assay reproducing menstrual phases was performed to determine LGR5 expression throughout the menstrual cycle. After tissue digestion, samples were centrifuged for 5 min at 1200 rpm, and single stromal cells were resuspended with SCM and placed in p100 plates. The next day, media was changed, and cells were grown to confluence. Primary cultures were passaged no more than twice to preserve integrity of the eSF, and 2 × 10^5^ cells were cultured in p6 well plates with SCM containing 2% FBS. Cells were treated with 10^−8^ M estrogen (E2) β-estradiol (Sigma, Saint Louis, MO, USA) for six days and with 10^−8^ M E2 and 10^−6^ M progesterone (P4) (Sigma) from days 6–12; a second group was treated only with ethanol as a vehicle, as both hormones were diluted in ethanol. Every two days, media was changed and hormones were added. Samples were obtained at days 0 (control), 6 (proliferative phase), and 12 (secretory phase). The experiment was performed in duplicate with 84 wells total. After treatment, cells were harvested using trypsin, and 106 cells were collected in 350 µL of lysis buffer with 1% β-mercaptoethanol and stored at −80 °C for subsequent RNA extraction.

### 4.4. RNA Extraction

RNA from 84 samples from the in vitro study described above was isolated using an RNeasy micro kit (Qiagen, Valencia, CA, USA) following manufacturer’s instructions, and concentration of RNA was determined using a Nanodrop^®^ photometer (Thermo-Fisher Scientific, Waltham, MA, USA). 

### 4.5. Real Time Quantitative PCR (RT-qPCR)

RT-qPCR of specific markers for the proliferative phase (CYR61) and secretory phase (DKK1) was performed. LGR5 expression was also studied to determine its variation throughout the menstrual cycle. Primer pairs are shown in Table 2. A total of 84 samples were reverse transcribed into complementary DNA (cDNA) using the SuperScript (Invitrogen) synthesis system, and RT-qPCR was performed using SYBR-green (Roche Life Sciences, Basel, Switzerland). Samples were analyzed in triplicate. GAPDH was used as a reference gene, and all data were normalized to its levels. Experimental data were compared to untreated cells as controls and were analyzed using one-way or two-way ANOVA followed by a Bonferroni comparison test, with *p* ≤ 0.05 indicating statistical significance. 

### 4.6. Immunocytochemistry and Fluorescence Activated Cell Sorting (FACS)

Eutopic endometria from 12 healthy women and 25 women with endometriosis were stained and cells were sorted using a BD FACS-ARIA I instrument (BD Bioscience, San Jose, CA, USA). We also stained 13 ectopic endometria from women with endometriosis. To elucidate whether LGR5 varied across the menstrual cycle, we performed a trial with five samples in the proliferative phase and four samples in the secretory phase. After tissue digestion, samples were treated with erythrocyte lysis buffer and blocked with 5% BSA for 1 hour at room temperature. Samples stained with monoclonal rabbit anti-LGR5 primary antibody (1 µL per million cells was used; BioNova Scientific, Fremont, CA, USA) and goat Alexa647 anti-rabbit secondary antibody (Invitrogen) at a 1:500 dilution. To discard dead cells, samples were stained with DAPI (5 mg/mL). LGR5^+/−^ cells were collected separately in TRIzol (Invitrogen) and stored at −80 °C. To confirm that LGR5^+^ cells were specifically sorted, we performed a cytospin of 5000 cells on a slide and stained them with the same antibody and concentration used for immunofluorescence (anti-LGR5/Gpr49 (loop 2) rabbit, Abgent) (Appendix A). Percentage of LGR5^+^ cells was analyzed by FCS Express5.0 software (De Novo Software, Glendale, CA, USA), and data analysis was performed using Prism software (GraphPad, San Diego, CA, USA). Two-way ANOVA and a Bonferroni post-test (*p* ≤ 0.05 indicated statistical significance) was used to find significant differences between phases of the menstrual cycle and between different types of endometriosis. A *t*-test (*p* ≤ 0.05) was used to determine differences between eutopic endometrium in control and endometriosis samples and between eutopic and matched ectopic endometrium in women with endometriosis. 

## Figures and Tables

**Figure 1 ijms-20-00022-f001:**
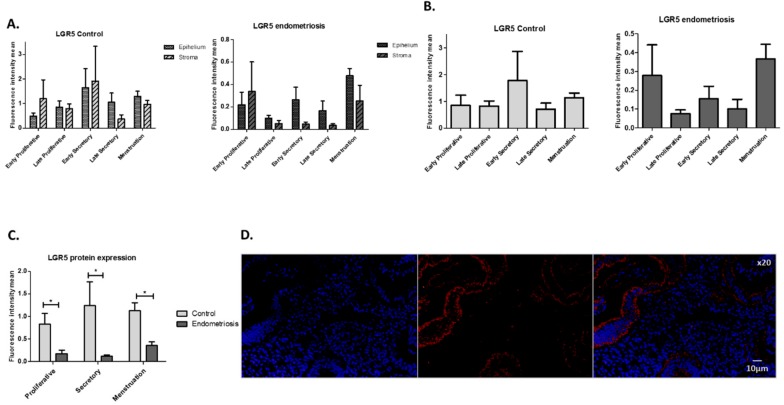
Immunofluorescence measurement of LGR5 expression throughout the menstrual cycle in eutopic endometrium. (**A**) Epithelial and stromal expression of LGR5 in control and endometriosis tissue in five phases of the menstrual cycle. (**B**) Total tissue expression across five phases of the menstrual cycle in control and endometriosis tissues. (**C**) Differences in LGR5 expression throughout the menstrual cycle between control and endometriosis tissue groups (*t*-test for each phase; proliferative: *p* = 0.0242; secretory: *p* = 0.0424; menstruation: *p* = 0.0121). (Control: *n* = 24; endometriosis: *n* = 24). (**D**) Example of immunofluorescence of LGR5 in eutopic endometrium (early secretory phase). In blue: DAPI; in red: LGR5. * *p* < 0.05.

**Figure 2 ijms-20-00022-f002:**
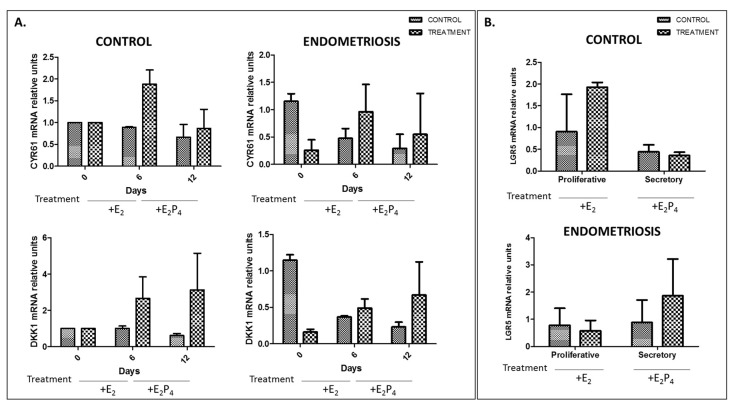
In vitro analysis of LGR5 expression throughout the menstrual cycle in endometrial stromal fibroblast primary culture after treatment (E_2_ and E_2_P_4_ for six and six more days, respectively). (**A**) Left panels show expression of CYR61 (marker of the proliferative phase) and DKK1 (marker of the secretory phase) in the control group (*n* = 4); right panels show expression of CYR61 and DKK1 in the endometriosis group (*n* = 3). (**B**) Upper panel shows expression of LGR5 in proliferative (E_2_) and secretory (E_2_P_4_) phases of the control group; lower panel shows LGR5 expression in both phases of the endometriosis group (E_2_: estradiol; P_4_: progesterone).

**Figure 3 ijms-20-00022-f003:**
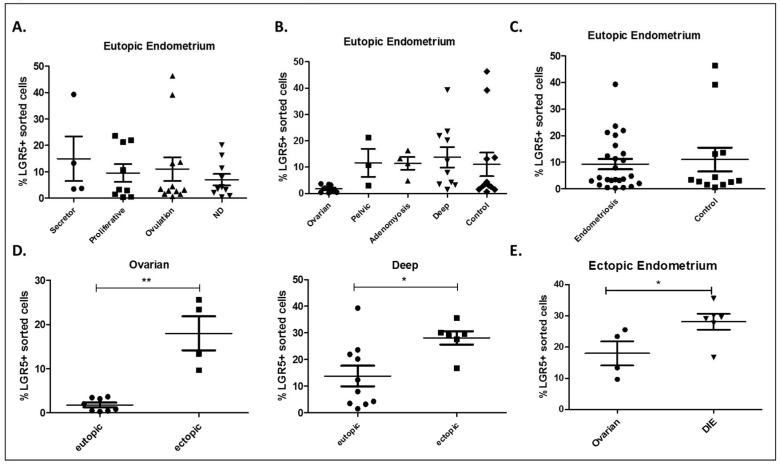
Percentage of FACS-sorted LGR5^+^ cells. (**A**) Percentages of LGR5^+^ cells in different phases of the cycle (proliferative: *n* = 9; secretory: *n* = 4; ovulatory: *n* = 12; ND: *n* = 9). (**B**) Percentage of LGR5^+^ cells in eutopic endometrium of different types of endometriosis and controls (ovarian: *n* = 8; pelvic: *n* = 3; adenomyosis: *n* = 4; DIE: *n* = 10; control: *n* = 12). (**C**) Percentage of LGR5^+^ cells identified in eutopic endometrium of women with and without endometriosis (control: *n* = 12; endometriosis: *n* = 25). (**D**) Differences in LGR5^+^ cells between eutopic and ectopic endometrium. Left panel: ovarian endometriosis (eutopic: *n* = 8; ectopic: *n* = 4; *p* = 0.0286). Right panel: DIE (eutopic: *n* = 10; ectopic: *n* = 6; *p* = 0.0411). (**E**) Difference between ovarian and DIE ectopic endometrium (ovarian: *n* = 4; DIE: *n* = 6; *p* = 0.0381). ND: non-determined; DIE: deep infiltrating endometriosis. * *p* < 0.05, ** *p* < 0.03.

**Table 1 ijms-20-00022-t001:** Patients used in the study.

Patient ID	Type of Patient	Type of Endometriosis	Eutopic Endometrium	Mached Ectopic Endometrium	Technique Used
**1**	Control	-	yes	-	FC
**2**	Control	-	yes	-	FC
**3**	Control	-	yes	-	FC
**4**	Control	-	yes	-	FC
**5**	Control	-	yes	-	FC
**6**	Control	-	yes	-	FC
**7**	Control	-	yes	-	FC
**8**	Control	-	yes	-	FC
**9**	Control	-	yes	-	FC/CC
**10**	Control	-	yes	-	FC/CC
**11**	Control	-	yes	-	FC/CC
**12**	Control	-	yes	-	FC/CC
**13**	Control	-	yes	-	IF
**14**	Control	-	yes	-	IF
**15**	Control	-	yes	-	IF
**16**	Control	-	yes	-	IF
**17**	Control	-	yes	-	IF
**18**	Control	-	yes	-	IF
**19**	Control	-	yes	-	IF
**20**	Control	-	yes	-	IF
**21**	Control	-	yes	-	IF
**22**	Control	-	yes	-	IF
**23**	Control	-	yes	-	IF
**24**	Control	-	yes	-	IF
**25**	Control	-	yes	-	IF
**26**	Control	-	yes	-	IF
**27**	Control	-	yes	-	IF
**28**	Control	-	yes	-	IF
**29**	Control	-	yes	-	IF
**30**	Control	-	yes	-	IF
**31**	Control	-	yes	-	IF
**32**	Control	-	yes	-	IF
**33**	Control	-	yes	-	IF
**34**	Control	-	yes	-	IF
**35**	Control	-	yes	-	IF
**36**	Control	-	yes	-	IF
**37**	Endometriosis	Ovarian	yes	-	FC
**38**	Endometriosis	Ovarian	yes	-	FC
**39**	Endometriosis	Ovarian	yes	-	FC
**40**	Endometriosis	Ovarian	yes	-	FC
**41**	Endometriosis	Ovarian	yes	yes	FC
**42**	Endometriosis	Ovarian	yes	yes	FC
**43**	Endometriosis	Ovarian	yes	yes	FC
**44**	Endometriosis	Ovarian	yes	yes	FC
**45**	Endometriosis	DIE	yes	yes	FC
**46**	Endometriosis	DIE	yes	yes	FC
**47**	Endometriosis	DIE	yes	yes	FC
**48**	Endometriosis	DIE	yes	yes	FC
**49**	Endometriosis	DIE	yes	yes	FC
**50**	Endometriosis	DIE	yes	yes	FC
**51**	Endometriosis	DIE	yes	-	FC
**52**	Endometriosis	DIE	yes	-	FC
**53**	Endometriosis	DIE	yes	-	FC
**54**	Endometriosis	DIE	yes	-	FC
**55**	Endometriosis	Pelvic	yes	yes	FC
**56**	Endometriosis	Pelvic	yes	-	FC
**57**	Endometriosis	Pelvic	yes	-	FC
**58**	Endometriosis	Adenomyosis	yes	-	FC
**59**	Endometriosis	Adenomyosis	yes	yes	FC
**60**	Endometriosis	Adenomyosis	yes	-	FC
**61**	Endometriosis	Adenomyosis	yes	yes	FC
**62**	Endometriosis	DIE	yes	-	CC
**63**	Endometriosis	DIE	yes	-	CC
**64**	Endometriosis	DIE	yes	-	CC
**65**	Endometriosis	ND	yes	-	IF
**66**	Endometriosis	ND	yes	-	IF
**67**	Endometriosis	ND	yes	-	IF
**68**	Endometriosis	ND	yes	-	IF
**69**	Endometriosis	ND	yes	-	IF
**70**	Endometriosis	ND	yes	-	IF
**71**	Endometriosis	ND	yes	-	IF
**72**	Endometriosis	ND	yes	-	IF
**73**	Endometriosis	ND	yes	-	IF
**74**	Endometriosis	ND	yes	-	IF
**75**	Endometriosis	ND	yes	-	IF
**76**	Endometriosis	ND	yes	-	IF
**77**	Endometriosis	ND	yes	-	IF
**78**	Endometriosis	ND	yes	-	IF
**79**	Endometriosis	ND	yes	-	IF
**80**	Endometriosis	ND	yes	-	IF
**81**	Endometriosis	ND	yes	-	IF
**82**	Endometriosis	ND	yes	-	IF
**83**	Endometriosis	ND	yes	-	IF
**84**	Endometriosis	ND	yes	-	IF
**85**	Endometriosis	ND	yes	-	IF
**86**	Endometriosis	ND	yes	-	IF
**87**	Endometriosis	ND	yes	-	IF
**88**	Endometriosis	ND	yes	-	IF

FC: flow cytometry; CC: cell culture; IF: immunofluorescence; ND: non-determined.

**Table 2 ijms-20-00022-t002:** Primers used for RT-qPCR.

Gene	Primer Name	Sequence 5′–3′	Tm
*CYR61*	hCYR61-For-25	CTCGCCTTAGTCGTCACCC	57.6
	hCYR61-Rev-226	CGCCGAAGTTGCATTCCAG	57.1
*DKK1*	hDKK1-For507	ATAGCACCTTGGATGGGTATTCC	56.6
	hDKK1-Rev-560	CTGATGACCGGAGACAAACAG	55.5
*LGR5*	hLGR5-For-71	CACCTCCTACCTAGACCTCAGT	57
	hLGR5-Rev-274	CGCAAGACGTAACTCCTCCAG	57.5
*GAPDH*	hGAPDH-For	CGT CTT CAC CAC CAT GGA GA	61.1
	hGAPDH-Rev	CGG CCA TCA CGC CAC AGT TT	56.7

Forward (For) and reverse (Rev) primers used for RT-qPCR.

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
