# Peer review of "Lgr5 Does Not Vary Throughout the Menstrual Cycle in Endometriotic Human Eutopic Endometrium"

_ijms, 2018, doi:10.3390/ijms20010022_

Round 1
Reviewer 1 Report
This study evaluates the expresison of the stem cell marker LGR5 in endometriosis and during the menstrual cycle uwing three different methodologies. As stem cells may be implicated in the pathogenesis of endometriosis due to their properties of unlimited proliferation and high developmental plasticity, the study follows a good rationale. This study represents an extension of previous findings of the authors (Fertil Steril 2017). The authors do not find a menstrual cycle-dependent expression of LGR5 in the endometrium of endometriosis patients, however, they can demonstrate increased expresison in ectopic endometrium vs eutopic endometrium. Conversely, the LGr5 expression is lower in eutopic endometrium compared to healthy endometrium. Inmvitro data demonstrate differential hormone-dependent regulation of stemness-related markers. Overall, the data contribute to a better characterization of stem cell factors in the endometrium and support the stem cell hypothesis of endometriosis. The manuscript is largely well-written, the methodology is solid, the patient collective is well-characterized and ethical approvcal is in place.
The manuscript could benefit from minor revisions, as outlined below:
- abstract: page 1 line 31..LGR being higher in ectopic lesions - please add 'compared to eutopic endometrium' because this is the comparison that was made.
- Page 2, line 77: I would find it useful to add a reference to the patient data here, e.g. "atient characteristics are shown in Table I". Also please mention briefly which three methods were used.
- Figure 1: As immunofluorescence data have been analysed, it would be nice to show an example picture of an immunostaining of LGR5 for visual impact and to allow qualitative assessment of the LGR5 staining.
- page 3, line 9.2 ff: The in vitro studies need to be introduced better in the results section and in Figure2: Please explain what the purpose of the in vitro study was and how it was performed (I know it is described in the methods section, but it needs to be explained to the reader also in the results). Explain the abbreviation eSF. Also, explain the study better in the Figure and legend (Fig.2). Fig.2 states 'TREATMENT', but it is not clear which treatment was performed. finally, no statistical analysis is provided for Figure 2. If the data were not significant, please mention this and dicsuss the data more clearly/carefully. Why were Cyr61 and Dkk1 studied (and not other markers). Was this choice based on previous work (e.g. the screening performed in Fertil Steril 2017 by the group)? Please exlpain.
- page 4, line 109. Please add an introductory sentence to the flow cytometry section to increase understandability. ('We next performed flow cytometric analysis in dissociated tissue to determine the percentage of LGR5 positive cells' or something similar). The text starts very abruptly, which may confuse the readers.
- Discussion: Figure 2 shows (non.significant?) changes in Cyr61 and DKK1. The discussion does not mention this result at all, please mention this result and discuss it.
Author Response
This study evaluates the expresison of the stem cell marker LGR5 in endometriosis and during the menstrual cycle uwing three different methodologies. As stem cells may be implicated in the pathogenesis of endometriosis due to their properties of unlimited proliferation and high developmental plasticity, the study follows a good rationale. This study represents an extension of previous findings of the authors (Fertil Steril 2017). The authors do not find a menstrual cycle-dependent expression of LGR5 in the endometrium of endometriosis patients, however, they can demonstrate increased expresison in ectopic endometrium vs eutopic endometrium. Conversely, the LGr5 expression is lower in eutopic endometrium compared to healthy endometrium. In vitro data demonstrate differential hormone-dependent regulation of stemness-related markers. Overall, the data contribute to a better characterization of stem cell factors in the endometrium and support the stem cell hypothesis of endometriosis. The manuscript is largely well-written, the methodology is solid, the patient collective is well-characterized and ethical approvcal is in place.
The manuscript could benefit from minor revisions, as outlined below:
- abstract: page 1 line 31..LGR being higher in ectopic lesions - please add 'compared to eutopic endometrium' because this is the comparison that was made.
Response1: We have added the suggested sentence in the abstract.
- Page 2, line 77: I would find it useful to add a reference to the patient data here, e.g. "patient characteristics are shown in Table I". Also please mention briefly which three methods were used
Response 2: We have added a sentence where we mention that we used endometrial biopsies and ectopic lesions of patients in Table1. We also mentioned the three methods used to determine if the marker was varying across the cycle.
- Figure 1: As immunofluorescence data have been analysed, it would be nice to show an example picture of an immunostaining of LGR5 for visual impact and to allow qualitative assessment of the LGR5 staining.
Response 3: We appreciate the comments and we have added the suggested changes in Figure 1.
- page 3, line 9.2 ff: The in vitro studies need to be introduced better in the results section and in Figure2: Please explain what the purpose of the in vitro study was and how it was performed (I know it is described in the methods section, but it needs to be explained to the reader also in the results).
Response 4: We explained better the purpose of the in vitro study in this section and a brief description of the methods used (see line 99).
Explain the abbreviation eSF.
Response 5: The abbreviation corresponds to endometrial Stromal Fibroblasts (eSF) and we have modified in the manuscript.
Also, explain the study better in the Figure and legend (Fig.2). Fig.2 states 'TREATMENT', but it is not clear which treatment was performed.
Response 6: We have performed the suggested changes in Figure 2.
Finally, no statistical analysis is provided for Figure 2. If the data were not significant, please mention this and dicsuss the data more clearly/carefully.
Response 7: We have performed the changes and inserted them in the discussion (see line 210).
Why were Cyr61 and Dkk1 studied (and not other markers).
Response 8: In the study, we also studied other markers, such as TNC for proliferative phase and Col4A for the secretory phase. However, the probes for the RT-qPCR, resulted to be highly unspecific, so we decided to use only those two markers.
Was this choice based on previous work (e.g. the screening performed in Fertil Steril 2017 by the group)? Please exlpain.
Response 9: In our previous work we performed a previous in silico study where we found that those genes were varying across the menstrual cycle in normal human endometrium. We have added this information in the manuscript (see line 212).
- page 4, line 109. Please add an introductory sentence to the flow cytometry section to increase understandability. ('We next performed flow cytometric analysis in dissociated tissue to determine the percentage of LGR5 positive cells' or something similar). The text starts very abruptly, which may confuse the readers.
Response 10: We appreciate the comments and we have modified this section adding an introductory sentence.
- Discussion: Figure 2 shows (non.significant?) changes in Cyr61 and DKK1. The discussion does not mention this result at all, please mention this result and discuss it.
Response 11: We have added this suggestion in the discussion.
Reviewer 2 Report
In the current paper the authors adressed whether LGR5 expression differs in several phases of the menstrual cycle.
Overall, this is a well written paper which needs several modifications.
Fig1 needs to be accompanied by respective fluorescence picures and the data need to be normalized to a houskeeping factor. To justify that the LGR levels do not change the data need to include one positive control undergoing respective changes.
Fig 2 the in vitro cell data need statistics. In addition, the "treatment "needs to be clearly indicated, not as treatment in the figure.
Author Response
In the current paper the authors adressed whether LGR5 expression differs in several phases of the menstrual cycle.Overall, this is a well written paper which needs several modifications.
Fig1 needs to be accompanied by respective fluorescence picures
Response 1: We appreciate the suggestions and we have added an image to Figure1.
And the data need to be normalized to a houskeeping factor.
Response 2: In our study, we decided not to use a housekeeping factor since our experiments were controlled with positive and negative controls. As a positive control, we used intestine embedded in paraffin, since LGR5 marker has been largely described in this tissue and it is found in the base crypts of this tissue. As negative controls, we always used the same tissue without any staining (neither primary nor secondary antibodies) and the same tissue stained only with the secondary antibody (to determine the background of the secondary antibody). Using those controls, we set up the background of the staining and determined the wavelength where the primary antibody was positive.
To justify that the LGR5 levels do not change the data need to include one positive control undergoing respective changes.
Response 3: In this case, we were using paraffin-embedded tissues. Therefore, we have used the Noyes criteria (please see reference below) which has been largely considered to be the gold standard to date the endometrium. This slides were evaluated by an expert pathologist (Dr. Castellví) and selected the slides and determined the the phases of the menstrual cycle of each one.
Additionally we knew the last menstrual period (LMP) of all these samples and we could date very precisely the endometrium with these clinical features (LMP) as well as the anatomo-pathological dating according Noyes criteria. This is the reason why we consider that using other genes will not add much value to determine the menstrual cycle phase.
Reference: Noyes RW, Hertig AT, Rock J. Dating the endometrial biopsy. Am J Obstet Gynecol. 1975 May;122(2):262-3.
Fig 2 the in vitro cell data need statistics.
Response 4: As we explained in the results section and discussion, the results are not significant. However, and as suggested by the reviewer, we added a sentence in Figure2 legend where we mentioned that * = p<0.05. However, we did not add any asterisk in the figure because there are not significant p-values.
In addition, the "treatment "needs to be clearly indicated, not as treatment in the figure.
Response 5: We have added this information to Figure2 as suggested by the reviewer.
Round 2
Reviewer 1 Report
The authors have appropriately addressed my previous requests.
Reviewer 2 Report
The authors have addressed the comments.